# Metabolomic Insights into the Phytochemical Profiles and Seasonal Shifts of *Fucus serratus* and *F. vesiculosus* Harvested in Danish Coastal Waters (Aarhus Bay)—An Untargeted High-Resolution Mass-Spectrometry Approach

**DOI:** 10.3390/md23110417

**Published:** 2025-10-26

**Authors:** Mihai Victor Curtasu, Jørgen Ulrik Graudal Levinsen, Annette Bruhn, Mette Olaf Nielsen, Natalja P. Nørskov

**Affiliations:** 1Department of Animal and Veterinary Sciences, AU Campus Viborg, Research Centre Foulum, Aarhus University, DK-8830 Tjele, Denmark; 2Department of Ecoscience, Aarhus University, DK-8800 Aarhus, Denmarknatalja.norskov@anivet.au.dk (N.P.N.); 3Centre for Circular Bioeconomy (CBIO), AU Campus Viborg, Research Centre Foulum, Aarhus University, DK-8830 Tjele, Denmark

**Keywords:** Fucales, brown seaweed, metabolomics, profiling, temporal shifts, clustering

## Abstract

This study investigated the year-round metabolomic variation in *Fucus serratus* (FS) and *F. vesiculosus* (FV) collected monthly from Danish coastal water around Aarhus Bay. Untargeted high-resolution liquid chromatography–mass spectrometry profiling (LC-HRMS), combined with multivariate data analysis and temporal clustering analysis, revealed that species identity was the primary driver of metabolic separation, followed by seasonal variation. FS showed higher levels of hydrolyzable tannins, flavonoid derivatives, aromatic amino acids, and glutamine-rich peptides, whereas FV was enriched in complex phlorotannins, tricarboxylic acid cycle intermediates, and carnitine derivatives. Temporal analysis identified recurring seasonal patterns across both species, including spring increases in amino acids, purine metabolites, and osmolytes; mid-summer peaks in mannitol and sulfated derivatives; and late-autumn elevations in phenolic compounds and betaine-type osmolytes. Despite apparent interspecific differences, several metabolite groups exhibited similar seasonal dynamics, suggesting shared physiological strategies associated with growth activation in spring, metabolic adjustment during summer to possible increased grazing pressure, and nutrient reallocation prior to winter. These findings provide a comprehensive, high-resolution view of seasonal metabolomic patterns in *Fucus* spp., offering new insights into their biochemical ecology and supporting the targeted utilization of these species for applications requiring specific metabolite profiles. Finally, this study contributes to the creation or expansion of metabolomic libraries for HRMS specific to *Fucus* seaweeds.

## 1. Introduction

Macroalgae are increasingly recognized as valuable biological resources with ecological, nutritional, and biotechnological significance. Two major species, *Fucus serratus* Linnaeus, 1753 and *F. vesiculosus* Linnaeus, 1753, dominate rocky intertidal zones across the North Atlantic [1,2,3]. Their rich phytochemical composition, characterized by polysaccharides, polyphenols, lipids, and secondary metabolites, has further attracted attention for applications in food, feed, pharmaceutical, and cosmeceutical industries [4]. This abundance of phytochemicals explains their extensive use across multiple fields. In human nutrition and health, *Fucus*-derived polysaccharides and polyphenols are investigated for antioxidant, anti-inflammatory, and antimicrobial properties, with promising applications in functional foods and nutraceuticals [5,6,7,8]. In animal nutrition, seaweed supplementation is explored for improving gut health, modulating rumen fermentation, and contributing to methane mitigation strategies in ruminants [9,10]. Beyond feed and food, *Fucus* metabolites are exploited in cosmetics formulations for skin protection and anti-aging effects, while alginates are indispensable in pharmaceutical and biomaterial industries [4]. Species of the genus *Fucus* are characterized by abundant polyphenolic compounds, mainly phlorotannins—polymers of phloroglucinol—responsible for their antioxidant and photoprotective functions [11]. Catarino et al. have reviewed in detail the phlorotannin composition in Fucales [11] and other phycochemical constituents of *Fucus* spp. [4]. Additionally, the ecological roles of these compounds, such as defense against herbivory and protection against ultraviolet radiation, highlight their adaptive significance and further emphasize the link between environmental seasonality and chemical variability [12].

There has also been an increased focus on using *Fucus serratus* and *F. vesiculosus* for methane reduction, given new insights into their biochemical composition [4,13]. Research shows that incorporating *F. vesiculosus* has the potential to reduce methane emissions when included in ruminant diets by affecting the rumen microbial community, which plays a crucial role in methane production during fermentation [10,14,15]. Furthermore, secondary bioactive metabolites present in seaweed may influence ruminal fermentation and methane production as previously found with in vitro studies [15,16,17,18]. The characterization of these metabolites, particularly how their profiles shift with seasonal changes, could shed light on the mechanisms underlying their antioxidant, antimicrobial, or even anti-methanogenic effects. Studies have shown that the timing of cultivation and harvest can enhance the efficacy of *Fucus* species in reducing methane emissions, suggesting that further investigation into their seasonal metabolomic profiles could yield important data to inform future interventions targeting enteric methane reduction [9,19]. For instance, the presence of phlorotannins and other polyphenolic compounds in brown algae such as *Fucus vesiculosus* could exert antimicrobial effects on key microbial populations responsible for methane production. However, the seasonal availability and concentration of these compounds can vary significantly [15,16], and they play a crucial role in determining the nutritional quality and availability of bioactive compounds within seaweed species [20,21], including *Fucus* species [19,22]. Research currently lacks understanding of the chemical variations throughout different seasons, which could further enhance their application in sustainable livestock production, serving both environmental and economic interests [10,23,24]. Unlike prior studies that primarily focused on general seasonal trends in selected metabolites or bioactivity endpoints, this study provides a high-resolution untargeted metabolomic analysis using dual-mode electrospray ionization mass spectrometry (ESI-MS), coupled with clustering and differential abundance models to explore both interspecific and intra-annual metabolic dynamics.

Understanding these dynamics would not only enrich our knowledge of the metabolic responses of *Fucus* species but also provide critical insights for optimizing their use in human nutrition and health, as well as in livestock as dietary supplements. The primary objective of this study was to gain a comprehensive understanding of the seasonal variations throughout a full year in the metabolome composition of two *Fucus* species, specifically *Fucus serratus* and *F. vesiculosus*, through monthly collections and untargeted exploration using high-resolution mass spectrometry metabolomics analysis.

## 2. Results and Discussion

### 2.1. Environmental Conditions

Overall, a strong seasonal fluctuation in key environmental parameters was observed where dissolved inorganic nitrogen (DIN) concentrations in seawater were highest during winter months, reaching values around 10 µM and 5 µM in January and February, respectively, and declined markedly towards summer, where concentrations stabilized below 1 µM (Figure 1A). In contrast, dissolved inorganic phosphorus (DIP) in seawater showed relatively low values throughout most of the year but exhibited a pronounced increase in late autumn and early winter, peaking at nearly 2 µM in December (Figure 1A). Temperatures ranged between 5 and 20 °C, increasing in May, and peaking in June to September, followed by a decline until minimum temperatures were reached in December (Figure 1B). Salinity remained relatively stable during most of the year (20–25 psu) but showed a modest decrease during spring and early summer (Figure 1B). Solar radiation exhibited the expected seasonal cycle, with low levels during November to February winter months, a gradual increase from March, and maximum daily values exceeding 300–400 W m^−2^ d^−1^ during June, followed by a decline towards autumn and winter (Figure 1C).

### 2.2. Proximate Composition and Polyphenol Content in Seaweeds

The dry matter (DM) content (% of FW) after freeze-drying and ash (% of DM) of FS and FV are presented in Figure 2. Stable content was observed for both species with slight variations according to the month of collection. The DM content for FS varied between 21.6% (April) and 29.4% (August), whereas the DM content for FV varied between 21.35% (May) and 31.1% (December). The variation in ash content (% of DM) throughout the 2023 season was similar to that of DM content, except for collections performed in October and November, when FV was highly overgrown with sinistral spiral tubeworms (*Spirorbis* sp. Linnaeus, 1758), resulting in a higher content of ash (Figure 2). The growth of *Spirorbis* on FS was not as pronounced as on FV. *Spirorbis* and related species are known to adhere externally to seaweed thalli and secrete rigid, coiled tubes composed predominantly of Mg-calcite [25]. Increased colonization on the seaweed surface can lead to decreased photosynthetic rates and growth of *Fucus* species [26]. The higher content of Mg-calcite, which leads to a high ash fraction in the harvested seaweed biomass, may lead to a mineral imbalance if the *Fucus* biomass is utilized as animal feed or soil fertilizer and can also decrease the proportion of fermentable organic compounds, like polysaccharides and bioactive molecules [27].

Total polyphenol content (TPC) was determined using the Folin–Ciocalteu assay, and results were expressed as mg/g equivalents of gallic acid (GAE) (Figure 3). Throughout the year of sampling, FS had a significantly higher concentration of polyphenols as compared to FV. TPC in FV peaked in the summer months of June and July, with the maximum analyzed value in August (23.3 mg/g equiv GAE). In contrast, for FS, the TPC peaked earlier in the year, corresponding to the months of April and May (36.98 and 37.07 mg/g equiv GAE), followed by a second increase in TPC in August (34.03 mg/g equiv GAE). The seasonal peaks in polyphenol content likely reflected increased oxidative stress or UV exposure during spring and summer, aligning with previously reported photoprotective roles of phlorotannins in brown algae [11,12]. The TPC patterns revealed clear seasonal dynamics, with FV showing a pronounced summer maximum (August) and FS exhibiting broader fluctuations across late spring–summer. Similar phlorotannin seasonality has been reported for Fucus spp. along temperate Atlantic coasts [28], often attributed to increased UV exposure and oxidative stress during periods of high irradiance [29]. Solar radiation observed for our location was maximal in May-June, corresponding to higher TPC values observed for FS. In another year-round survey employing pressurized liquid extraction and comprehensive metabolomic profiling, maximal 2,2-diphenyl-1-picrylhydrazyl (DPPH) scavenging and Ferric Reducing Antioxidant Power (FRAP) antioxidant capacities were observed in late spring and early summer, concomitant with peaks in total phenolic content, which was attributed primarily to phlorotannins [30]. We also observed that, although individual samples were collected from the same area, a large degree of variation was observed at different time points, both for FS and FV, indicating a wide range of individual responses within a closed population. However, it must be acknowledged that the increased intra-species variation observed monthly might also reflect the Folin–Ciocalteu TPC method and its sensitivity to other compounds, as any co-extracted reducing substances (e.g., carbohydrates, proteins, or inorganic reductants) can contribute to the apparent TPC and thus potentially lead to an overestimate of the actual polyphenol concentrations [31].

### 2.3. Untargeted Metabolomics Profiling—Principal Component Analysis Overview

Untargeted metabolomics profiling was performed on the FS and FV samples collected in 2023 to understand the metabolome activity at a species level, as well as how it is modulated by environmental seasonality. Data was collected in both positive ionization mode (ESI+) and negative ionization mode (ESI−). Using multivariate data analysis, the ESI+ features and ESI− were subjected to principal component analysis (PCA), as shown in Figure 4 and Figure 5, respectively. For both datasets, the PCA score plot revealed a significant separation between FS and FV, regardless of the collection time. The separation in the direction of PC1 was also responsible for the highest amount of variation recorded, 37.2% in PC1 for ESI+ (Figure 4) and 36% in PC1 for ESI− (Figure 5). Secondary PC and separation of samples in this direction were explained by 23.2% and 24.7% variations for the ESI+ and ESI− datasets, respectively. It was observed that separation according to the month of sampling is occurring for both FS and FV in the direction of PC2. Both FS and FV exhibit similar clustering patterns for January, February, March, and April, indicating a higher degree of separation between these monthly recordings and the rest of the months. FS and FV samples collected from May are quite distinguishable in both ESI+ and ESI−, indicating a higher degree of dissimilarity between May and the rest of the months. Furthermore, another distinct cluster can be observed for samples collected in June, July, August, and September, indicating that samples in these months share a high degree of similarity. Finally, while October, November, and December appear to overlap with other samples, they form a distinct cluster that can be better visualized when examining PC3 as having its own direction in 3D space (Appendix A). This observed cyclical pattern indicates that there are seasonally structured metabolomic changes that likely coincide with photoperiod, temperature, or life cycle stage. The temperature and day length increased in May to peak in June/July, with relatively high temperatures (>15 °C) remaining until September. A previous study on metabolic profiling of the surface metabolome by GC-MS has also reported substantial seasonal differences between spring/summer and autumn/winter [12]. Despite their close taxonomic relationship, FS and FV displayed distinct metabolomic signatures throughout the year. This divergence likely reflects species-specific adaptations to their respective intertidal microhabitats as FV experiences more pronounced emersion and photic stress. Although Rickert et al. (2016) described the surface metabolite composition in both FS and FV, they mainly focused on seasonal differences and not the direct comparison of FS and FV [12]. To the best of our knowledge, this is the first time an untargeted metabolomics study examines the differences between FS and FV in seaweed biomass using LC-HRMS. Overall, these findings confirm that both seasonality and species identity have a substantial effect on the metabolomic composition of *Fucus* spp. and highlight the importance of considering sampling time when interpreting ecological or functional metabolomics data from seaweeds.

### 2.4. Species-Specific Metabolomic Differences

The metabolic features responsible for discriminating between FS and FV, as determined by the comparative analysis model, are illustrated in Figure 6. Information on all compounds, including unknown compounds and ontology classification, can be found in Appendix A.

FS showed an enriched expression in polyphenolic compounds, dihydrocaffeic acid 3-sulfate (log_2_FC = +4.15), dihydroferulic acid 4-O-sulfate (+4.04), and two unidentified hydrolyzable tannins (ID284, ID307), along with quercetin pyruvate (+3.04). These compounds, and generally polyphenols, are associated with antioxidant defense, UV protection, and chemical defense mechanisms [11,12,32]. Under high UV exposure, polyphenols accumulate in the surface layers of *Fucus* spp., absorbing destructive UV-B before it can generate phototoxic radicals. In FV, Creis et al. demonstrated that UV-B irradiation triggers both constitutive and inducible increases in phlorotannin content alongside upregulated expression of key phenolic biosynthesis genes, representing a dual-layered photoprotective strategy [32]. Rickert et al. (2016) further correlated seasonal peaks in surface polyphenol concentrations with maximal incident solar radiation and minimum epiphytic fouling, suggesting that FS exploitation of UV-screening polyphenols may balance the need for photoprotection and antifouling [12]. A correlation analysis between the metabolome profile and TPC was performed and is presented in Appendix A. There were several metabolites positively correlated with TPC, including two very strong correlations of dihydrocaffeic acid 3-sulfate and dihydroferulic acid 4-O-sulfate (*p* < 0.001 and rho > 0.65). These compounds are derivatives of the hydroxycinnamic acids caffeic and ferulic acid, which are well-known phenolic compounds [33]. On the other hand, FV showed significantly higher concentrations of phlorotannins, such as fucodiphloroethol G and its isomer (−6.34 and −6.71), diphlorethohydroxycarmalol (−4.19), and trifucol (−4.70), indicating different phlorotannin expressions between species despite both being members of the *Fucus* genus. It has been previously shown that FV contains a greater absolute concentration of polymeric phlorotannins than FS, which correlates with reduced microbial fouling and a distinct epiphytic community composition on its thallus surface [11,34]. Rickert et al. further showed that seasonal peaks in FV phlorotannin content coincide with maximal UV fluxes and minimal epibiont colonization, implying that FV’s heavier phlorotannin oligomers serve dual roles in photoprotection and antifouling [12]. In this regard, it is noticeable that FS, the *Fucus* species with the highest concentrations of polyphenols in this study, had the lowest occurrence of visible biofouling from *Spiropsis* sp. of the two species investigated. Interestingly, phlorotannin-like compounds were strongly negatively correlated to TPC (Appendix A), indicating that the total polyphenol assay may not fully capture their abundance, possibly due to differences in assay reactivity or structural features that limit detection by the Folin–Ciocalteu method. Therefore, despite FV containing higher levels of complex phlorotannins based on untargeted profiling, its TPC values measured by the Folin–Ciocalteu method were consistently lower. This discrepancy likely reflects limitations of the TPC assay, which preferentially detects monomeric or low-molecular-weight phenolics with strong reducing capacity, while underestimating high-molecular-weight or heavily polymerized phlorotannins characteristic of FV. Consequently, the TPC absolute values reported here likely underestimate the true phlorotannin content. Additionally, sulfation and structural complexity may further reduce assay reactivity. Seasonal harvesting strategies, guided by the metabolic patterns and differences between *Fucus* species, can maximize yields of target phlorotannin profiles, offering a path toward sustainable valorization of coastal resources into high-value bioactive compounds [30].

Regarding amino acid metabolism, several amino acids and dipeptides are more abundant in FS, including glutamic acid (+1.37), phenylalanine (+0.67), tyrosine (+0.83), tryptophan (+0.67), and glutamine-containing di- and tripeptides (Gln-Gln-Gln, Pyr-Gln-Gln). These differences could potentially indicate an elevated amino acid turnover or nitrogen assimilation in FS. In brown macroalgae, glutamic acid and its amide derivative glutamine constitute the principal carriers of assimilated inorganic nitrogen, shuttling the reduced nitrogen through the classical glutamine synthetase/glutamate synthase (GS/GOGAT) cycle and distributing it between uptake sites and growth or storage tissues. It has been previously demonstrated that in *Fucus vesiculosus* glutamate and glutamine are dynamically distributed across thallus tissues, reflecting active nitrogen assimilation and translocation processes [35]. The observed enrichment of these compounds in FS could suggest an upregulated GS/GOGAT flux and enhanced amino acid turnover relative to FV. Significantly higher tissue content of N has also been observed in FV as compared to FS [36]. Moreover, the observed increase in aromatic amino acids, including phenylalanine, tyrosine, and tryptophan, suggests an increased activity of the shikimate pathway, which funnels aromatic precursors into the biosynthesis of polyphenolic and flavonoid structures [37]. Conversely, adenosine (−1.27), carnitine (−2.67), and its derivative hydroxyoctadecadienylcarnitine (−1.32) were enriched in FV, suggesting a potential shift toward purine metabolism and β-oxidation. GSSG (oxidized glutathione; +0.74) was found in higher abundance in FS, indicating a potentially more oxidizing cellular environment or increased redox cycling compared to FV. For organic acids and carbohydrate-related metabolites, FV was enriched in aconitic acid (−2.55), isocitric acid lactone (−5.11), and 2-furoic acid and its isomer (−1.31 and −0.86), many of which are intermediates in the TCA cycle or degradation products of sugars. Meanwhile, FS showed elevated levels of maltitol (+1.08), which may reflect differences in osmolyte accumulation or carbon storage strategies. Several sulfated compounds showed specificity towards FS, which had more dihydrocaffeic acid 3-sulfate and choline sulfate, whereas FV was enriched in 2-ethylsulfanylpropanoate and 2-(3-hydroxypropylamino)ethanesulfonic acid. This could indicate either a different sulfur metabolism or enhanced detoxification strategies specific to FS. Similar sulfated compounds, for example, dimethylsulfopropionate (DMSP), have been previously reported in FV with antibacterial attachment properties, indicating that seaweeds are capable of producing a diverse range of ecologically relevant chemical compounds that act as surface inhibitors against bacterial-driven biofouling [38].

### 2.5. Temporal Variation of the Fucus Metabolic Profiles

Apart from the species differences, we were able to follow the progression of metabolite intensity during the monthly collections and could observe temporal differences for specific metabolites. The results of the time-course clustering analysis using the untargeted metabolomics profiles of FS and FV are shown in Figure 7. To explore time-dependent shifts in metabolic profiles, soft clustering analysis was performed using the Mfuzz algorithm. Six temporal clusters were identified based on standardized metabolite abundance patterns (Z-scores) across the 12-month sampling period for both FS and FV. We observed that four clusters (Cluster 1, Cluster 2, Cluster 4, and Cluster 5) exhibited similar metabolite patterns over time for FS and FV, whereas Cluster 3 contained metabolites with opposite activities. Specifically, FS metabolites in cluster 3 showed increased activity in January, February, March, and April, followed by a gradual decrease in the summer-autumn months. On the other hand, FV metabolites in Cluster 3 showed increased activity in the summer months and a decrease towards late autumn and December.

In Cluster 1, both species exhibit a gradual rise in metabolite levels towards the autumn/winter months. A shared temporal trend is observed across species, with metabolite abundance peaking primarily in November, as shown in the heatmap of metabolic features shared between FS and FV in Cluster 1 (Figure 8). Notably, this cluster includes annotated compounds such as dihydrocaffeic acid 3-sulfate and 4-hydroxyphenylacetic acid sulfate, both of which are derivatives of phenolics known for their antioxidant activity and roles in cell wall fortification. Their seasonal accumulation in late autumn may reflect enhanced oxidative stress protection during colder months or a response to photoperiod reduction. Given this increase later in the year, it is unlikely that these phenolics are a response to UV exposure. It has been previously shown that arylsulfotransferases, specifically ast6, which are involved in sulfate transfer and increasing water solubility and stabilization of phenolic compounds, were not increased with higher exposure to UV-B [32]. On the other hand, the presence of sulfated phenolics could be associated with an increased detoxification and excretory function seen after the summer months [39]. A higher presence of isocitric acid lactone and proline betaine was observed in FS and FV during October and November. These metabolites are known to be involved in central metabolism and osmolyte regulation, with potential roles in carbon flow and cellular homeostasis under seasonal stress. Proline betaine is a known osmoprotectant that could support the hypothesis of environmental stress adaptation mechanisms in autumn and early winter. Previous studies have reported the presence of betaines and other tertiary sulphonium compounds in marine macroalgae (e.g., glycinebetaine, gamma-aminobutyric acid betaine, proline betaine) with protective functions by contributing to osmotic regulation through lowering cellular osmotic potential and maintaining turgor under hyperosmotic conditions [40,41]. In the intertidal red alga *Pyropia haitanensis*, mild dehydration has been shown to increase the presence of free proline, while moderate to severe dehydration was seen to enhance the additional accumulation of betaine-type osmolytes (e.g., alanine betaine) and sugars, thereby sustaining water retention and turgor pressure [42]. Moreover, given the observed increase in *Spirorbis s.* biofouling specifically in October and November, it should not be excluded that these metabolites and others in Cluster 1 might be a physiological response to environmental stressors. Several unknown or partially annotated compounds with consistent seasonal accumulation across species were also identified (e.g., Unknown_ID589_322.0061, Unknown_ID250_375.0705, etc.), which may represent novel or understudied metabolites specific to *Fucus* spp. Cluster 4 displays an increase in the Spring-Summer months, followed by a decline, suggesting metabolites associated with mid-season physiological transitions such as storage compound accumulation or stress adaptation (Figure 7). Species differences can be observed in Cluster 4, as some metabolites from FS peak in July and September, compared to FV. Only two metabolic features were shared across Cluster 4 by both FS and FV; however, they are unfortunately not annotated features (Appendix A).

Cluster 2 captures metabolites with a mid-year peak in June–August (Figure 9). FV shows a more substantial and more defined increase compared to FS, indicating species-specific upregulation of summer-associated metabolites, particularly in June, July for FS and August for FV (Figure 7). We observed in Cluster 2 several annotated compounds, notably multiple mannitol-related features (e.g., mannitol [pos], [M + Cl]-mannitol, 2-deoxy-2-fluoro-D-glucitol) which exhibited peak abundance in mid-summer, particularly in FV (Figure 9).

Mannitol is a primary carbon storage compound in brown algae and acts also as an osmoprotectant, protecting cells from dehydration and high salinity [43,44]. Its increased abundance during warmer months likely reflects a combination of heightened photosynthetic activity, as also described from kelps [20] and osmotic adjustment, which mitigates water loss during tidal emersion [45]. FS and FV shared a similar trend in mannitol accumulation. However, the intensity of the peak was consistently higher in FV, suggesting species-specific regulation or metabolic allocation during peak biomass production. Additional features enriched in Cluster 2 include sulfated derivatives, such as 2-(sulfamoylamino)pentanedioic acid, 3-sulfopropanediol, and 2-(3-hydroxypropylamino)ethanesulfonic acid, which are potential osmolytes or intermediates in detoxification. As previously mentioned, with sulfated compound from cluster 1 and given the known activity of ast6 involved in sulfate transfer [32], it is unclear whether the increase in sulfated derivatives is a direct response to increased UV-B exposure or it is associated with an increased detoxification and excretory function during the summer months [39]. Several unknown metabolites (e.g., Unknown_ID26_317.0552_RT0.67) also followed the summer peak pattern. For example, *m*/*z* 317.0552 has been putatively annotated as another sulfated derivative ([6-(2,3-dihydroxypropoxy)-3,4,5-trihydroxyoxan-2-yl]methanesulfonic acid, C_9_H_18_O_10_S). Based on its chemical fragmentation pattern, it has been categorized as a monosaccharide structure using CANOPUS and ClassyFire. A high proportion of these compounds was mainly selected in negative ionization mode, indicating a very specific ionization and an early elution (<1.0 min) characteristic of very soluble compounds that poorly bind to a C18 column. However, the consistent temporal profiles across both species support their biological relevance despite lack of full annotation.

Cluster 5 contains metabolites that exhibit a declining temporal trend, with the highest abundance observed in January–March, followed by a gradual decrease throughout the rest of the year (Figure 10). This pattern is consistent across both FS and FV, although FS metabolites generally exhibit slightly higher relative abundances, particularly during the early months. Among the annotated compounds in this cluster, several are phenolic or flavonoid-like structures. These include quercetin pyruvate, a derivative of the flavonoid quercetin, and 5,6,7,8,2′,3′,5′-heptahydroxy-4′-methoxyflavanone, which may represent a polyhydroxylated flavonoid with strong antioxidant potential. An unidentified hydrolyzable tannin (ID307_513.0659) was also annotated. The abundance of these early-season metabolites may reflect the physiological need for antioxidant protection during winter and early spring, when the photoperiod increases but stress from cold or oxidative damage remains elevated. It has been previously reported that *Fucus spiralis* harvested in winter had higher concentrations of total phenolics and flavonoids, as well as antioxidant activities (DPPH-scavenging activity), which declined by spring [46]. Another annotated compound in this cluster is loliolide (Figure 10), a monoterpenoid lactone previously reported in seagrasses and seaweeds, such as *Sargassum ringgoldianum* subsp. *coreanum* [47] and *Undaria pinnatifida* [48]. Loliolide is associated with effects on cell protection and antioxidant activities [47], and its early-season peak may indicate involvement in developmental transitions and protection over the winter months. Additional unknown features, such as Unknown_ID436_237.1485 and Unknown_ID791_323.0405, follow similar trajectories and may represent early expressed secondary metabolites with uncharacterized roles in environmental sensing or metabolic regulation.

Although several metabolites were shared with similar trajectories in unique clusters, as presented earlier, some specific temporal clustering was evident for FS and FV (Figure 7). Notably, FS in Cluster 3 (Figure 11A) and FV in Cluster 6 (Figure 11B) displayed similar temporal profiles, marked by a pronounced increase in abundance during the early spring months (February–April), followed by a gradual decline through summer and early autumn, and a modest recovery towards late autumn and early winter (November–December). This parallel dynamic in metabolite composition and temporal shifts between FS Cluster 3 and FV Cluster 6 indicates that both species undergo comparable seasonal shifts in key metabolic pathways (Figure 11). Several shared metabolites belong to the amino acid metabolism network, including glutamic acid, tyrosine, and isoleucine, together with a range of glutamine-rich peptides (e.g., Gln-Gln-Ala, Gln-Gln-Phe, Gln-Leu-Gln, Gln-Gln-Asn, Pyr-Gln-Gln, and pyroglutamylglutamine). Phenylalanine and tryptophan were picked for clustering analysis only for FS, indicating a lower dynamic shift in FV. However, both these amino acids share the same aromatic biosynthesis pathway as tyrosine. A previous study on the dynamic distribution of metabolites in FV thallus has shown the presence of glutamic acid and phenylalanine, with glutamic acid higher in the apices and stipe, whereas phenylalanine was higher in apices and receptacles [35]. Generally, these compounds are central to nitrogen assimilation, protein synthesis, and protein turnover, and their elevated abundance during February–April is consistent with a period of increased nitrogen limitation and enhanced biosynthetic activity associated with growth and reproductive preparation. Although not many studies have reported on the seasonal shifts of amino acids in FS or FV, a recent study on *F. virsoides* has shown that exposure to herbicide glyphosate can impact the shikimate pathway of aromatic amino acid biosynthesis, which can lead to deregulations of protein synthesis, growth promoters, flavonoids or other phenolic compounds [49]. This is in line with the anticipated role of phenylalanine, tyrosine or tryptophan in the growth, development and protection of *Fucus* species. The pronounced summer decline in these metabolites likely reflects both reduced availability of inorganic N during summer, and a metabolic shift from growth towards maintenance, carbohydrate accumulation and reproduction with the increase in photosynthetic activity, while the modest increase in November–December may signal increased availability of inorganic N co-occurring with pre-winter nitrogen remobilization or storage [12].

A second group of features shared in both FS and FV is associated with purine metabolism, including adenine and, in FS, adenosine monophosphate. The seasonal pattern, with a maximum in spring and a decrease in summer, suggests that nucleotide turnover is tightly coupled to growth dynamics. Although direct metabolite profiling in *Fucus* is limited, transcriptomic evidence in FV demonstrates springtime upregulation of genes annotated to purine-nucleotide biosynthesis and turnover, consistent with heightened cell division and increased biomass [50]. In FV, oxidized glutathione (GSSG) followed a similar trend, pointing to a concurrent modulation of redox homeostasis during phases of high metabolic activity. Genes related to oxidative stress response have been previously shown to be upregulated in FV as a response to hyposalinity acclimation, including glutathione reductase [50]. Choline sulfate, detected in both species, is indicative of osmolyte accumulation and enhanced sulfur metabolism in spring. This enrichment could be linked to osmotic stress management as a response to seasonal changes in salinity.

Since this study was conducted over a single annual cycle, it must be recognized as a limitation that it only reflects a preliminary annual survey and may not capture interannual variability in metabolite profiles. Long-term monitoring across multiple years is necessary to discriminate between intra- versus interannual trends in *Fucus* spp. metabolomes.

## 3. Materials and Methods

### 3.1. Materials

The following chemicals were used for the extraction and analysis of seaweed: methanol of HPLC grade was purchased from Sigma (Merck KGaA, Darmstadt, Germany) and acetonitrile (ACN) was purchased from WVR (Søborg, Denmark). Formic acid (FA) of LC-MS grade was purchased from Fluka (Merck KGaA, Darmstadt, Germany). For LC-QTOF-MS, the following chemicals were used: lithium formate monohydrate 98% was purchased from Sigma (Merck KGaA, Darmstadt, Germany); isopropanol, acetonitrile, and formic acid, all LC-MS grade (VWR, West Chester, PA, USA); MILLIQ LC-MS grade water (Milli-Q^®^ IQ 7000 ultrapure water system coupled with a LC-Pak^®^ Polisher; Merck KGaA, Darmstadt, Germany).

### 3.2. Seaweed Collection

Seaweed material, *Fucus serratus* (FS) and *F. vesiculosus* (FV), was manually harvested every month from 29.01.2023 to 13.12.2023 in Begtrup Vig (56°16′ N 10°52′ E), a small inlet within Aarhus Bay, Denmark. All samples were collected from the same site, ensuring environmental consistency across the sampling campaign. Begtrup Vig is relatively shallow with 10–12 m depth in the deepest parts. The bottom is mostly sandy, interspersed with larger boulders, rocks, and pebbles, and the vegetation is dominated by Zostera marina, FS, and FV. Although it is generally a sheltered area, this western-facing shoreline is exposed to westerly winds, allowing waves to build up across 18 km of Aarhus Bay. Species determination was based on morphology, with both species having distinct morphological characteristics [51]. Five individual biological replicates of each species were collected in clear plastic bags, each containing a minimum of 50 g of wet biomass. The collection was carried out within the same area running along roughly 20 m of shoreline and 20 m perpendicular to the coastline. Harvesting was carried out by abscission, using scissors or a knife, of approximately 1/3 of each individual seaweed frond. Care was taken to include both old parts near the stipes as well as the youngest parts, the apical tips. The harvested individuals were collected based on being representative of the general population at the study site with regard to individual size and degree of fouling. The collected seaweeds were transported in thermocol boxes on the same day and stored at −20 °C until processing for analysis. Before freeze-drying, the seaweed material was placed at −80 °C for 24 h. The freeze-dryer ScanVac CoolSafe (LaboGene A/S, Lillerød, Denmark) operated at −40 °C for 72 h. After freeze-drying, the samples were placed into an exicator for 10 min before the final weighing. The freeze-dried material was then ground using a Retsch ZM200 Grinder–Gemini BV rotor grind mill (Retsch-Allee, Haan, Germany) and passed through a 0.5 mm particle size ring screen. Contents of dry matter (DM) (% of fresh weight (FW)) and ash were determined by oven-drying at 103 °C for 24 h, and ash content by combustion at 525 °C for 6 h.

### 3.3. Environmental Parameters and Chemical Analysis of the Seawater Nutrient Concentrations

Monthly point measurements of temperature and salinity were conducted at the study site, and 3 filtered water samples were collected for analyses of dissolved inorganic nutrient concentrations (nitrogen (N) and phosphorus (P)). Temperature (°C) was recorded using a digital thermometer (Ebro Handheld, TFX 410-1 Pt1000; Xylem Analytics Germany GmbH–EBRO, Ingolstadt, Germany). Salinity (psu) was measured using a refractometer (Bellingham & Stanley). The filtered seawater samples were analyzed for dissolved inorganic nitrogen (DIN) in the form of nitrate, nitrate and ammonium, and dissolved inorganic phosphorus (DIP) in the form of orthophosphate using a five-channel SKALAR San plus segmented flow autoanalyzer (Breda, The Netherlands). All methods were adopted from Hansen et al. (1999) [52]. The radiation data was acquired from the Danish Meteorological Institute (DMI) weather station Tranebjerg Øst (station number 05165) through DMI Open Data (Available online: https://opendatadocs.dmi.govcloud.dk/en/DMIOpenData, accessed on 19 September 2025). Radiation is presented as a daily mean with grey and standard error.

### 3.4. Analysis of Seaweed Material

Total Polyphenol content (TPC) was determined using the Folin–Ciocalteu assay, and sample preparation was done as previously described [53]. For untargeted metabolomics profiling, a sample preparation method was used based on methanol (100%) extraction. Firstly, 50 mg of seaweed material were weighed and added to 1.25 mL of methanol containing internal standards (ISs) for LC-MS analyses (*p*-chlorophenylalanine and glychocholic acid (glycine-1-13C)), then mixed and sonicated for 15 min and vortexed for 1 h at room temperature. The supernatant (200 µL) was diluted with 0.1% formic acid in MILLIQ water 1:3 supernatant/water and centrifuged for 10 min at 4 °C at 29,700× *g* and transferred to an HPLC glass vial. Extracts were analyzed by ultra-high pressure liquid chromatography (UHPLC) using a Nexera × 2 liquid chromatography system and an LCMS-9030 quadrupole time-of-flight mass spectrometry (Q-TOF MS) system (Shimadzu Corporation, Kyoto, Japan) in positive and negative electrospray ionization (ESI+/ESI−) mode [18]. Chromatographic separation was performed with an Acquity HSS T3 column (1.7 µm, 100 × 2.1 mm, Waters Ltd., Elstree, UK). The column temperature was set to 40 °C, and the samples were maintained at 18 °C in the autosampler. The injected sample volume was 5 µL, and the eluent flow was set to 400 µL/min. The chromatographic system used a binary gradient of Solvent A (water with 0.1% formic acid) and Solvent B (acetonitrile with 0.1% formic acid). The gradient started at 5% of solvent B pre-equilibration for 0.1 min, followed by a linear increase to 100% B over 20 min. The gradient followed a 2 min isocratic step at 100% solvent B. The gradient returned to 5% B followed by a post-equilibration period of the column for 3 min. The full chromatographic run was 25 min long. Data collection was performed in MS1 mode covering a 50–700 *m*/*z* range, and quality control (QC) samples were analyzed in data-dependent acquisition (DDA) mode covering a 50–700 *m*/*z* range, where 10 dependent events (precursor ions) were selected continuously for fragmentation during the run. A precursor intensity threshold was set at 1800, and precursor ions were fragmented with 20 eV collision energy (CE) with a ±10 eV CE spread. The following MS parameters were used: ion-source temperature, 300 °C; heated capillary temperature, 250 °C; heat block temperature, 400 °C; electrospray voltage 4.0 kV (ESI+) or −3.5 kV (ESI−); electrospray nebulization gas flow, 3 L/min; drying gas flow, 10 L/min; detector voltage, 2.02 kV and Ar was used as a collision gas for mass fragmentation. Mass calibration was performed externally using a sodium iodide solution (400 ppm in methanol) from *m*/*z* 50–1000. Data acquisition was performed using LabSolutions software version 5.96 (Shimadzu Corporation, Kyoto, Japan). Quality control samples (QCs) were prepared for all matrices by pooling aliquot amounts of all samples, using the same protocol as the samples. These were used to monitor the quality and stability of the chromatographic runs by injecting the QCs multiple times throughout the run in a series of five QCs. Two types of blanks were used to monitor external contamination or carryover effects: an ACN sample and a blank containing MILLIQ water and internal standards, which were prepared with the same procedure as the samples. Due to the high number of samples, they were separated into two different plates and ran as two consecutive batches.

### 3.5. Preprocessing of LC-QTOF-MS Data

The collected raw spectra files were converted to mzML format and used for preprocessing in MZmine version 4.7.9 by mzio GmbH [54,55]. The data extraction was performed using the mzwizard batch processing flow. Centroid mass detection was applied to both MS1 and MS2 levels. Parameters for the ADAP chromatogram builder, including the minimum number of consecutive scans, minimum intensity for consecutive scans, minimum peak intensity, and *m*/*z* (mass-to-charge ratio) tolerance, were carefully defined. The wavelet algorithm was used for the local minimum feature resolver, with optimized settings. Chromatograms were deisotoped using a 13C isotope filter, adjusting the *m*/*z* and retention time tolerances, with a maximum charge of 2 and a selection of the most intense isotopes as representatives. Additional steps included identifying isotopic peaks, aligning features with the join aligner, and gap filling for peak detection. The resulting feature list was exported as a .csv file containing RT, *m*/*z*, and peak area information. ESI+ data preprocessing resulted in a raw data matrix of 120 samples and 170 variables, whereas ESI- resulted in a data matrix of 120 samples and 279 variables. Information was also extracted for molecular networking (GNPS, FBMN) and SIRIUS. Compound annotation in mzmine was performed against publicly available databases (Available online: https://systemsomicslab.github.io/compms/msdial/main.html#MSP, accessed on 10 April 2025) and using SIRIUS+CSI:FingerID version 6.2.2 with the integrated tools of CSI:FingerID (with COSMIC), ZODIAC and CANOPUS/ClassyFire [56]. The levels of compound annotation have been described as previously suggested by Sumner et al. [57]. The positive and negative data sets were further processed to filter out low confidence filters based on missing values in the QCs and in biological samples (where missing values were found in more than 50% of the biological samples, the variable was excluded) and prepared for batch correction using QC:MXP version 2.0 [58]. Within-batch correction mode was performed using a spline algorithm, and between-batch correction was performed using the QC samples with a 3-fold cross-validation method and 10 Monte Carlo repetitions.

### 3.6. Multivariate Data Analyses

After batch correction and further filtration to eliminate unwanted adducts and known fragments (determined after the compound annotation step), the positive and negative datasets containing 79 and 90 variables, respectively, were used for data analysis. Prior to multivariate analysis, the metabolomics data matrix was log-transformed using log_2_(x + 1) to reduce heteroscedasticity. Pareto scaling was subsequently applied to each feature by centering and dividing by the square root of its standard deviation. Principal Component Analysis (PCA) was performed using the “prcomp()” function. Centroids for each species-month group were calculated as the mean PC1 and PC2 scores. The first two principal components (PC), as well as the first and third PC, were visualized using “ggplot2 4.0.0”, with individual samples coloured by month and shaped by species. Ellipses representing 68% confidence intervals were drawn around each month–species group to illustrate within-group variability. To identify metabolites that discriminate between FS and FV, a differential abundance analysis was performed between species using a linear modelling approach implemented with the “limma” package. The log-transformed data were used to construct a design matrix without an intercept, modelling the fixed effect of species. The model was fitted using “lmFit” followed by empirical Bayes moderation via “eBayes”. Differentially abundant features were extracted using the “topTable” function, and a Benjamini-Hochberg correction was applied for multiple testing. Features with an adjusted *p*-value < 0.05 and absolute log_2_ fold change > 0.5 were considered significant. Finally, for the characterization of seasonal patterns from the metabolomics profiles, a time-course clustering analysis using the “Mfuzz” package was performed [59]. Data from FS and FV were analyzed independently. The dataset was transposed, log_2_-transformed and standardized (Z-score) followed by conversion into an ExpressionSet object, with time-points assigned as phenotype data (months). The optimal fuzzifier parameter m was estimated using “mestimate()”, and soft clustering was performed with six clusters using the “mfuzz()” function. The number of clusters (c = 6) was determined based on cluster stability and biological interpretability. Features were assigned membership scores to each cluster, and those with membership ≥ 0.8 were considered high-confidence members of a cluster. Clustering was performed using Euclidean distance. These features were extracted and used for visualization. Clustering profiles that contained similar patterns were plotted together, allowing for comparison of the seasonal expression patterns across clusters. All analysis and figures output were performed in RStudio 2025.05.0 Build 496 “Mariposa Orchid” Release (Posit Software, PBC, Boston, MA, USA).

## 4. Conclusions

This study provides an explorative overview of a year-round untargeted metabolomic assessment of FS and FV collected in Nordic waters. Multivariate analysis revealed that species identity was a primary factor of separation, with seasonality second. Specific temporal phases were identified, including early-year metabolite enrichment, mid-summer peaks in carbon storage compounds such as mannitol, and late-autumn increases in phenolic derivatives and osmolytes. FS was enriched in hydrolyzable tannins, flavonoid derivatives, and certain sulfated phenolics, while FV contained higher levels of complex phlorotannins and selected TCA cycle intermediates. Amino acid metabolism also differed between species, with FS showing higher levels of glutamic acid, aromatic amino acids, and glutamine-rich peptides. Shared trends in temporal clustering indicate that despite taxonomic and metabolic distinctions, both species engage in common physiological programs linked to growth initiation in spring, metabolic downregulation during summer, and resource reallocation prior to winter. It is important to mention, as a limitation of this study, that the use of a C18 reverse-phase column provided broad coverage of semi-polar to non-polar metabolites, but likely underrepresented highly polar compounds (e.g., certain sugars, small organic acids, inorganic ions) and highly hydrophobic molecules. Moreover, structure annotation accuracy was limited by the lack of macroalgae-specific mass spectrometry databases. However, putative identifications were supported by fragmentation patterns and in silico tools. Dedicated spectral libraries for *Fucus* or brown algae would enhance metabolite coverage and reduce annotation uncertainty. The untargeted LC–HRMS dataset generated in this study captures extensive chemical diversity in *Fucus serratus* and *F. vesiculosus* and provides a valuable foundation for developing dedicated spectral libraries for brown algae. This data will help improve metabolite annotation confidence and support the continued expansion of reference databases for marine macroalgae. Although this study did not evaluate bioactivity parameters directly, the observed temporal patterns and metabolite identities provide a biochemical basis for future targeted evaluations of antioxidant or other bioactivity potential. Overall, this work establishes a metabolomic framework for understanding seasonal chemical variation in *Fucus* spp., providing valuable insights for optimizing harvest timing for bioactive compound yield and for applications such as methane mitigation in ruminant nutrition.

## Figures and Tables

**Figure 1 marinedrugs-23-00417-f001:**
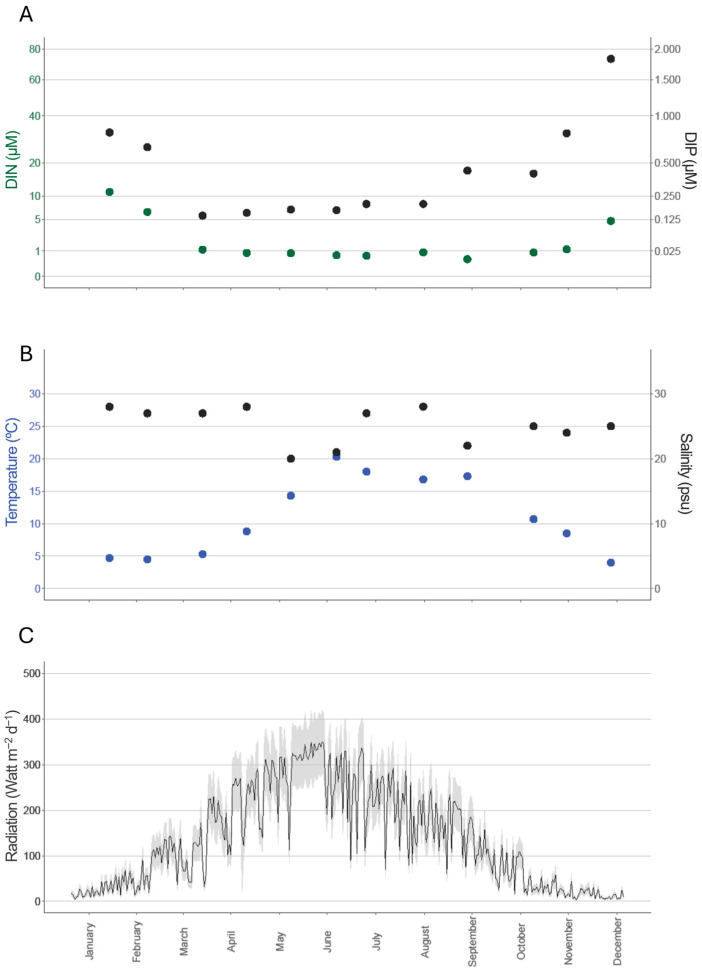
Seasonal variation in environmental parameters recorded at the sampling site: dissolved inorganic nitrogen (DIN, µM; green) and inorganic phosphorus (DIP, µM; black) in seawater (panel (**A**)); seawater temperature (°C; blue) and salinity (practical salinity units, black) (panel (**B**)); daily solar radiation (W m^−2^ d^−1^, mean ± SD) (pannel (**C**)). Data is shown across months from January to December.

**Figure 2 marinedrugs-23-00417-f002:**
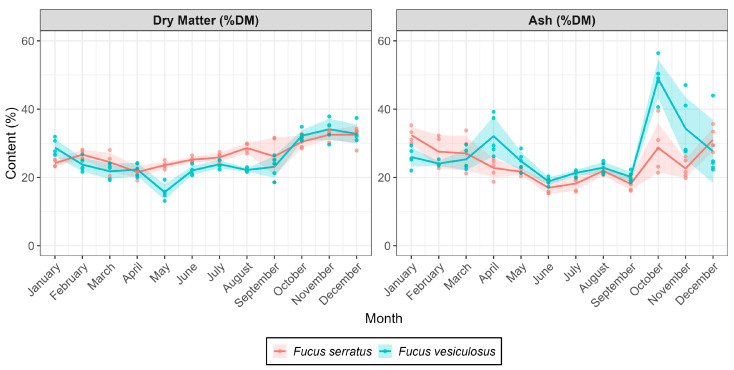
Dry matter content (% of FW) (after freeze drying) and ash (% of DM) content of *Fucus serratus* and *F. vesiculosus* during monthly collections.

**Figure 3 marinedrugs-23-00417-f003:**
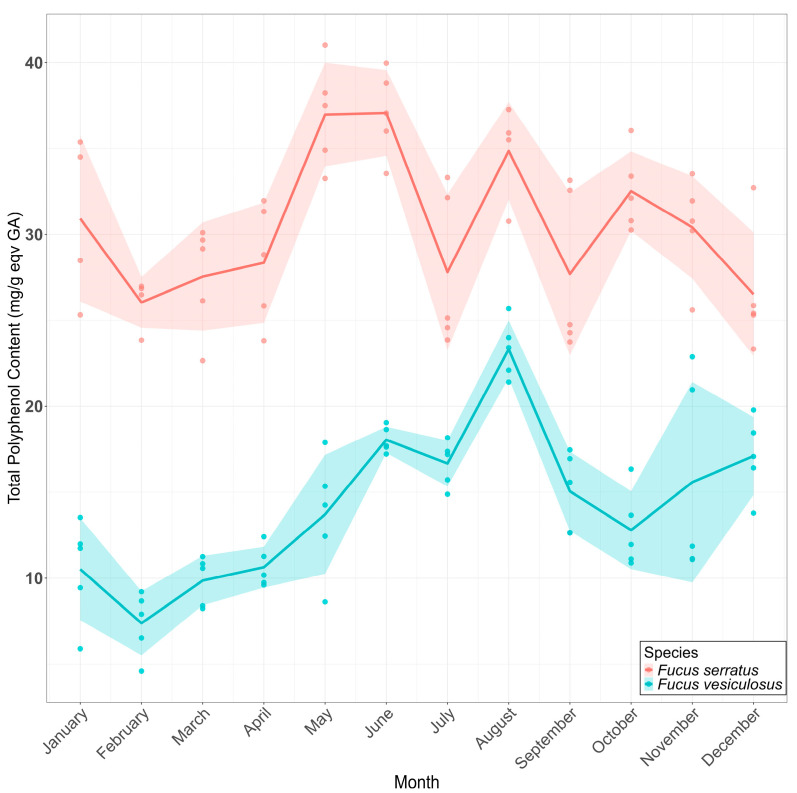
Total polyphenol content (TPC) in *Fucus serratus* and *F. vesiculosus* collected every month during a full year. Thick lines represent the average for each corresponding month. Data points represent independent biological replicates (*n* = 5 individual thalli), and the shaded areas represent the standard deviation from the mean.

**Figure 4 marinedrugs-23-00417-f004:**
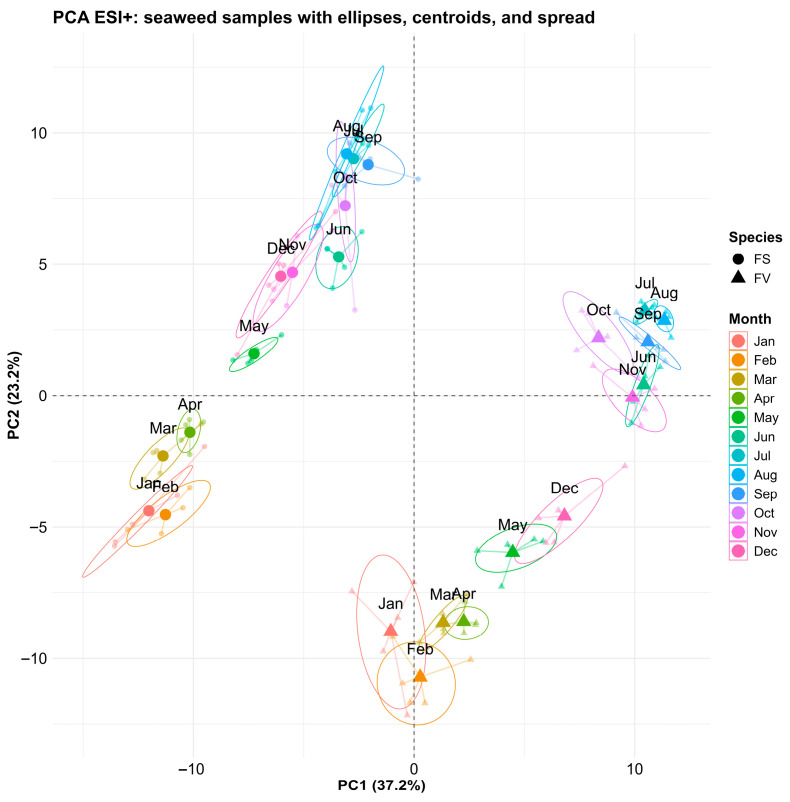
Principal component analysis (PCA) of untargeted metabolomic profiles (ESI+ LC-HRMS) of *Fucus serratus* (FS, circles) and *F. vesiculosus* (FV, triangles). Colored ellipses (95% confidence intervals) and centroids indicate intra-month variation and seasonal grouping.

**Figure 5 marinedrugs-23-00417-f005:**
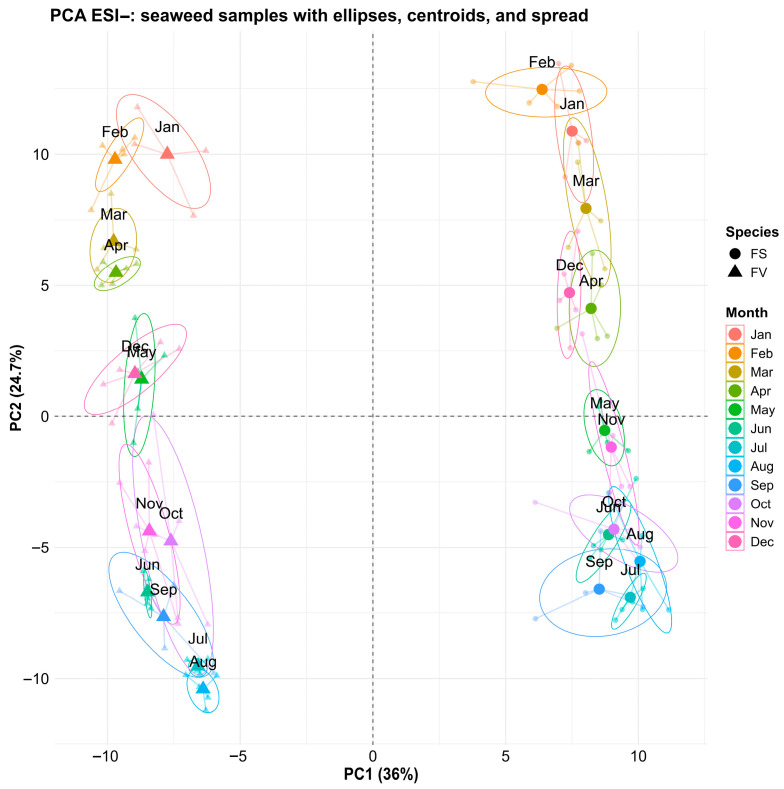
Principal component analysis (PCA) of untargeted metabolomic profiles (ESI− LC-HRMS) of *Fucus serratus* (FS, circles) and *F. vesiculosus* (FV, triangles). Colored ellipses (95% confidence intervals) and centroids indicate intra-month variation and seasonal grouping.

**Figure 6 marinedrugs-23-00417-f006:**
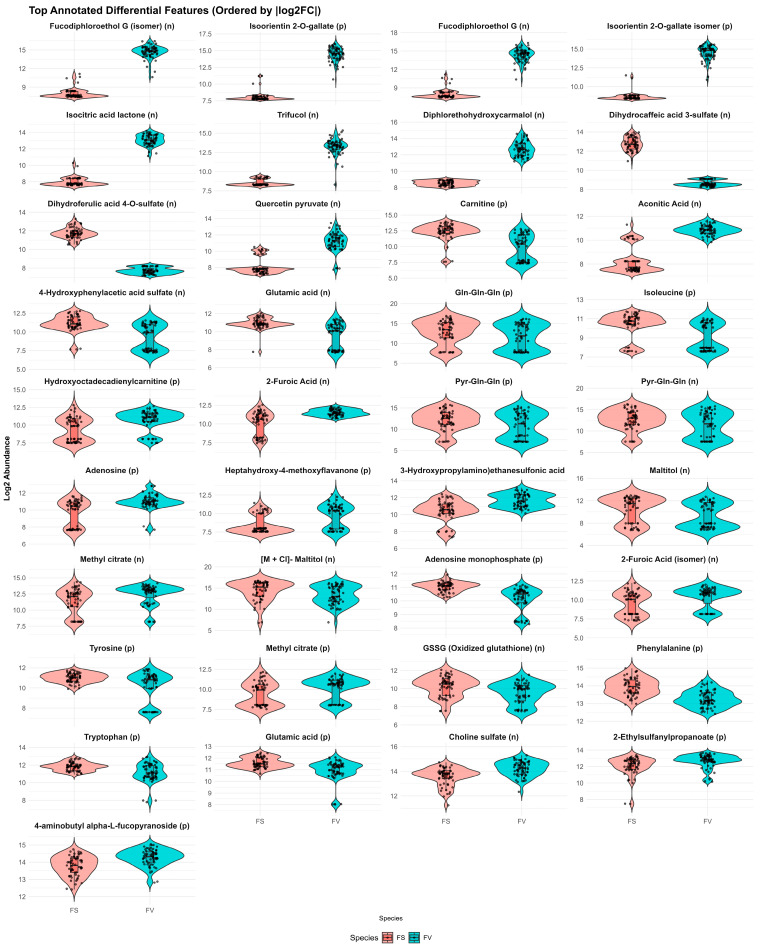
Top differentially abundant metabolites discriminating between *Fucus serratus* (FS) and *F. vesiculosus* (FV) detected in positive (*p*) and negative (*n*) ESI modes. Each plot represents the log_2_-transformed abundance of individual LC–HRMS features, ordered by absolute log_2_ fold change (log_2_FC) between species.

**Figure 7 marinedrugs-23-00417-f007:**
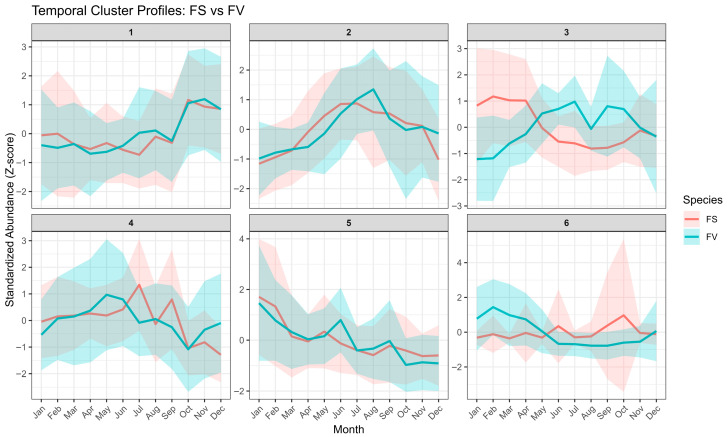
Temporal clustering of metabolomic features from *Fucus serratus* (FS, red) and *F. vesiculosus* (FV, blue) across 12 sampling months using Mfuzz soft clustering. Each panel (clusters 1–6) shows the standardized abundance (Z-score) profile of metabolites grouped by similarity in temporal trends. Solid lines indicate mean standardized abundance, and shaded areas represent the 95% confidence interval across features within each cluster.

**Figure 8 marinedrugs-23-00417-f008:**
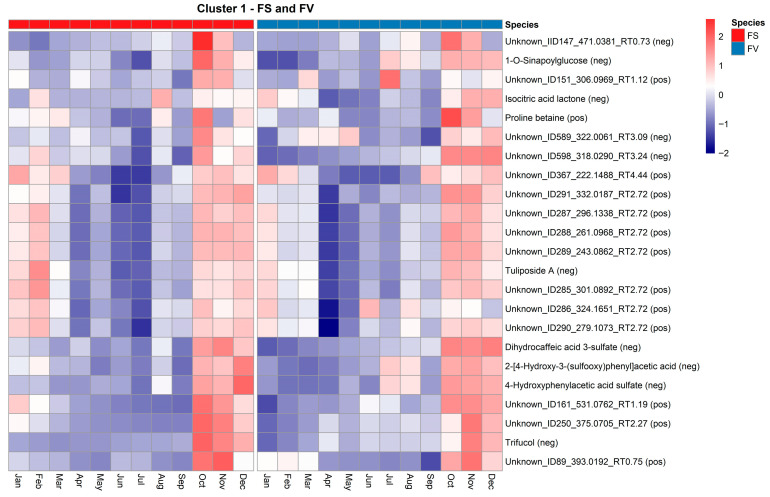
Heatmap of standardized abundance (Z-score) for metabolites in Cluster 1 shared by *Fucus serratus* (FS, red bar) and *F. vesiculosus* (FV, blue bar). Each cell represents the standardized abundance (z-score) of a metabolite, with red indicating higher relative abundance and blue indicating lower relative abundance within each species. The z-scores range from −2 (lowest) to +2 (highest), based on normalized LC–HRMS intensity values.

**Figure 9 marinedrugs-23-00417-f009:**
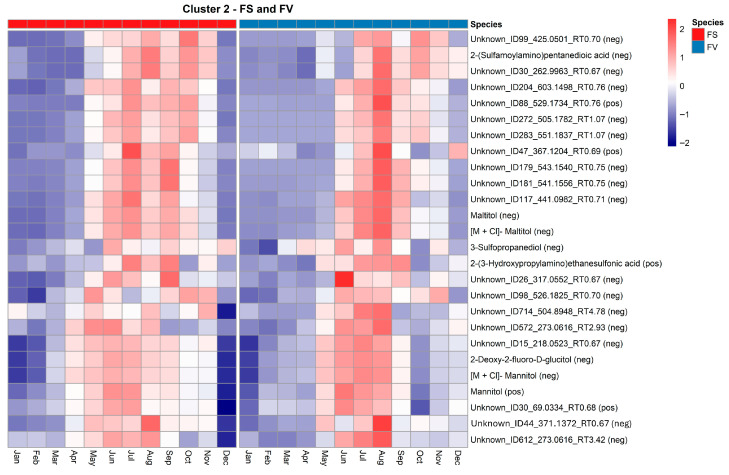
Heatmap of standardized abundance (Z-score) for metabolites in Cluster 2 in *Fucus serratus* (FS, red bar) and *F. vesiculosus* (FV, blue bar). Each cell represents the standardized abundance (z-score) of a metabolite, with red indicating higher relative abundance and blue indicating lower relative abundance within each species. The z-scores range from −2 (lowest) to +2 (highest), based on normalized LC–HRMS intensity values.

**Figure 10 marinedrugs-23-00417-f010:**
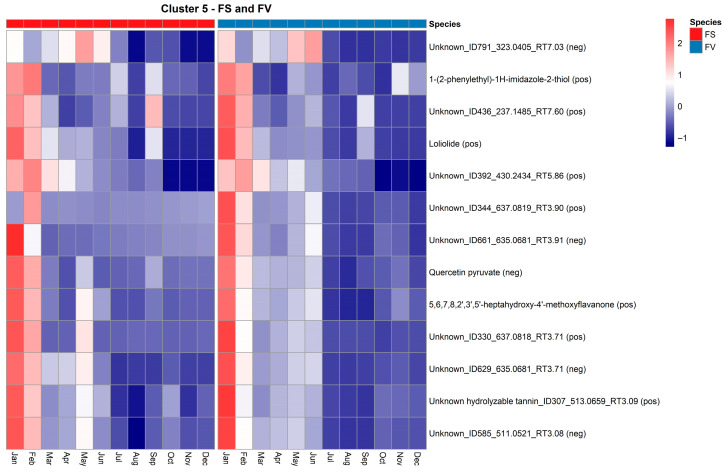
Heatmap of standardized abundance (Z-score) for metabolites in Cluster 5 in *Fucus serratus* (FS, red bar) and *F. vesiculosus* (FV, blue bar). Each cell represents the standardized abundance (z-score) of a metabolite, with red indicating higher relative abundance and blue indicating lower relative abundance within each species. The z-scores range from −2 (lowest) to +2 (highest), based on normalized LC–HRMS intensity values.

**Figure 11 marinedrugs-23-00417-f011:**
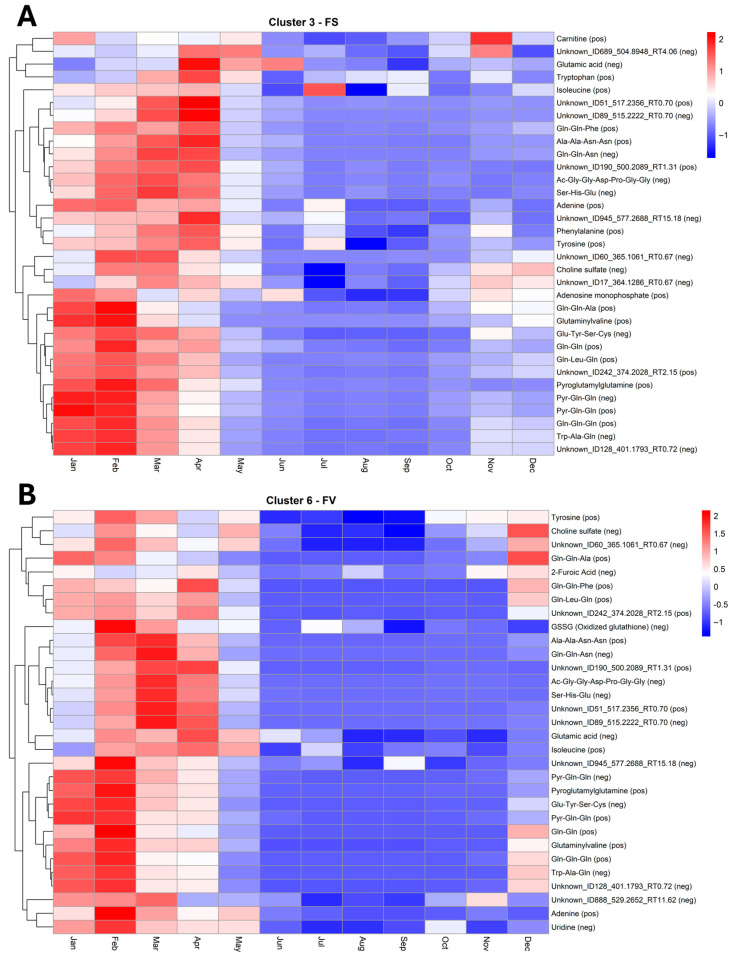
Heatmap of standardized abundance (Z-score) for metabolites in Cluster 3 of *Fucus serratus* (FS, panel (**A**)) and Cluster 6 of *F. vesiculosus* (FV, panel (**B**)). Each cell represents the standardized abundance (z-score) of a metabolite, with red indicating higher relative abundance and blue indicating lower relative abundance within each species. The z-scores range from −2 (lowest) to +2 (highest), based on normalized LC–HRMS intensity values.

## Data Availability

Dataset and metadata in connection to the study is available at https://www.ebi.ac.uk/metabolights (accessed on 23 September 2025), under accession number: MTBLS13204 (available online: https://www.ebi.ac.uk/metabolights/MTBLS13204, accessed on 23 September 2025).

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
