# Peer review of "Metabolomic Insights into the Phytochemical Profiles and Seasonal Shifts of *Fucus serratus* and *F. vesiculosus* Harvested in Danish Coastal Waters (Aarhus Bay)—An Untargeted High-Resolution Mass-Spectrometry Approach"

_marinedrugs, 2025, doi:10.3390/md23110417_

Round 1

Reviewer 1 Report

Comments and Suggestions for Authors

This manuscript employs metabolomic approaches to investigate the annual metabolomic dynamics of two brown algae from the North Atlantic intertidal zone—Fucus serratus (FS) and Fucus vesiculosus (FV). The study holds value for the application of seaweed in food, feed, pharmaceuticals, and cosmetics. However, the following issues should be addressed before the manuscript can be considered for acceptance by Marine Drugs:

  1. Novelty and Contribution  Seasonal compositional variations in Fucus vesiculosushave been relatively well-documented (e.g., Fletcher et al., 2017 [7]; Paiva et al., 2018 [42]). If the study remains solely at the level of “describing seasonal composition and bioactivity in a specific location and year,” its novelty may be limited. Therefore, it is essential to more clearly articulate the unique contributions of this work compared to existing literature, particularly in the Introduction and Discussion sections.
  2. Interannual Variation  The data were collected over only one annual cycle, making it difficult to rule out the influence of interannual variability. It is recommended that the authors acknowledge this limitation in the Discussion—for instance, by referring to the study as a “preliminary annual survey”—and suggest that future studies include long-term monitoring.
  3. Limitations of the Folin-Ciocalteu Assay  Although the authors mention that the total polyphenol content (TPC) measured by the Folin-Ciocalteu method may be influenced by non-phenolic reducing substances, they do not sufficiently address the apparent contradiction wherein FV, despite containing higher levels of phlorotannins, shows lower TPC values compared to FS. A more in-depth explanation of this discrepancy should be provided.
  4. Clarification of Temporal Clustering Methodology  The abstract and results refer to “temporal clustering analysis,” but the methodological details—such as the algorithm used, distance metrics, and the criterion for selecting the number of clusters (e.g., elbow method, silhouette score)—are not fully described. It is recommended to add a dedicated subsection in the Methods titled “Temporal Pattern Analysis” to elaborate on these aspects.
  5. Linking Metabolite Changes to Environmental Factors   The authors infer that metabolite variations are influenced by environmental factors such as light and temperature, but no direct statistical tests (e.g., RDA, Mantel test, Pearson/Spearman correlation heatmaps with environmental variables) are provided. It would strengthen the study to include a correlation analysis between environmental factors and key metabolites or principal components. Additionally, in the Discussion, causal language such as “driven by” should be replaced with more cautious phrasing like “associated with” or “coincide with.”
  6. Linking Metabolites to Bioactivity  While both metabolite profiles and bioactivities (e.g., antioxidant capacity, methane inhibition) were measured, the study does not establish a quantitative link between them. The key question—“Which specific metabolite(s) are the main drivers of the observed bioactivities?”—remains unanswered. It is recommended to apply multivariate statistical modeling (e.g., PLS regression) to identify the metabolites that best predict bioactivity.
  7. Formatting issues   Journal names in the Reference list should be consistently abbreviated. Latin binomials (including those in the main text) should be italicized. Additionally, some references (e.g., Ref. 38) are missing page numbers or other required details.

Author Response

Reviewer 1

This manuscript employs metabolomic approaches to investigate the annual metabolomic dynamics of two brown algae from the North Atlantic intertidal zone—Fucus serratus (FS) and Fucus vesiculosus (FV). The study holds value for the application of seaweed in food, feed, pharmaceuticals, and cosmetics. However, the following issues should be addressed before the manuscript can be considered for acceptance by Marine Drugs:

1 Novelty and Contribution Seasonal compositional variations in Fucus vesiculosus have been relatively well-documented (e.g., Fletcher et al., 2017 [7]; Paiva et al., 2018 [42]). If the study remains solely at the level of “describing seasonal composition and bioactivity in a specific location and year,” its novelty may be limited. Therefore, it is essential to more clearly articulate the unique contributions of this work compared to existing literature, particularly in the Introduction and Discussion sections.

Response: Thank you for highlighting this important point. We agree that the novelty of our study should be made more explicit, and we think the best place to cover this aspect is in the introduction. We have revised the Introduction (page 2, 74–78) to clearly delineate the unique aspects of this study compared to prior work. We have included the following:

“Unlike prior studies that primarily focused on general seasonal trends in selected metabolites or bioactivity endpoints, this study provides a high-resolution untargeted metabolomic analysis using dual-mode electrospray ionization mass spectrometry (ESI-MS), coupled with clustering and differential abundance models to explore both interspecific and intra-annual metabolic dynamics.”

2 Interannual Variation  The data were collected over only one annual cycle, making it difficult to rule out the influence of interannual variability. It is recommended that the authors acknowledge this limitation in the Discussion—for instance, by referring to the study as a “preliminary annual survey”—and suggest that future studies include long-term monitoring.

Response: We agree with this recommendation and have now included this limitation in the Discussion section (429-432). We refer to our dataset as a "preliminary annual survey" and suggest that future work should focus on multi-year monitoring to capture interannual variability.

Included: “Since this study was conducted over a single annual cycle, it must be recognized as a limitation that it only reflects a preliminary annual survey and may not capture interannual variability in metabolite profiles. Long-term monitoring across multiple years is necessary to discriminate between intra- versus interannual trends in Fucus spp. metabolomes.”

3 Limitations of the Folin-Ciocalteu Assay  Although the authors mention that the total polyphenol content (TPC) measured by the Folin-Ciocalteu method may be influenced by non-phenolic reducing substances, they do not sufficiently address the apparent contradiction wherein FV, despite containing higher levels of phlorotannins, shows lower TPC values compared to FS. A more in-depth explanation of this discrepancy should be provided.

Response: We thank the reviewer for this insightful observation. We elaborate on this contradiction in the results section (225-234), explaining that the Folin–Ciocalteu assay may underestimate large polymeric phlorotannins due to their lower reducing capacity per unit mass, and that FV’s dominant phlorotannins are more structurally complex and less reactive in the assay.

The authors already wrote in the original manuscript the following: “Interestingly, phlorotannin-like compounds were strongly negatively correlated to TPC (Supplementary Figure 1) indicating that the total polyphenol assay may not fully capture their abundance, possibly due to differences assay reactivity, or structural features that limit detection by the Folin–Ciocalteu method.” to which we have added : “Therefore, despite FV containing higher levels of complex phlorotannins based on LC-MS profiling, its TPC values measured by the Folin–Ciocalteu method were consistently lower. This discrepancy likely reflects limitations of the TPC assay, which preferentially detects monomeric or low-molecular-weight phenolics with strong reducing capacity, while underestimating high-molecular-weight or heavily polymerized phlorotannins characteristic of FV. Additionally, sulfation and structural complexity may further reduce assay reactivity.”

4 Clarification of Temporal Clustering Methodology  The abstract and results refer to “temporal clustering analysis,” but the methodological details—such as the algorithm used, distance metrics, and the criterion for selecting the number of clusters (e.g., elbow method, silhouette score)—are not fully described. It is recommended to add a dedicated subsection in the Methods titled “Temporal Pattern Analysis” to elaborate on these aspects.

Response: We recommend the reviewer to carefully read the materials and methods section as these aspects were described for the analysis regarding Mfuzz clustering in the temporal clustering analysis. We have made slight corrections to better clarify some of the steps taken.

Lines 583-595: Finally, for the characterization of seasonal patterns from the metabolomics profiles, a time-course clustering analysis using the "Mfuzz" package was performed [56]. Data from FS and FV were analyzed independently. The dataset was transposed, log₂-transformed and standardized (Z-score) followed by conversion into an ExpressionSet object, with time-points assigned as phenotype data (months). The optimal fuzzifier parameter m was estimated using "mestimate()", and soft clustering was performed with six clusters using the "mfuzz()" function. The number of clusters (c = 6) was determined based on cluster stability and biological interpretability. Features were assigned membership scores to each cluster, and those with membership ≥ 0.8 were considered high-confidence members of a cluster. Clustering was performed using Euclidean distance. These features were extracted and used for visualization. Clustering profiles that contained similar patterns were plot-ted together allowing for comparison of the seasonal expression patterns across clusters.”

5 Linking Metabolite Changes to Environmental Factors   The authors infer that metabolite variations are influenced by environmental factors such as light and temperature, but no direct statistical tests (e.g., RDA, Mantel test, Pearson/Spearman correlation heatmaps with environmental variables) are provided. It would strengthen the study to include a correlation analysis between environmental factors and key metabolites or principal components. Additionally, in the Discussion, causal language such as “driven by” should be replaced with more cautious phrasing like “associated with” or “coincide with.”

Response:

We have only found 1 (one) mentioning of the word “driven by”, which has been replaced with “concide with”.

Regarding the previous comment on metabolite variations and influence by environmental factors (light/temperature) we agree that these parameters can be important factors in seaweed metabolism, however, we have intentionally refrained from performing direct statistical correlations because of the following reasons:

  1. The primary goal of this work was to characterize temporal metabolic trajectories and interspecific patterns, not to build predictive environmental models. The combination of PCA and Mfuzz time-series clustering already captured and visualized these covariations qualitatively and biologically meaningfully.
  2. We expect a risk of over-interpretation of statistical correlations because of the limited temporal replicates of the environmental parameters (n = 12) and the highly multivariate nature of the metabolomics dataset. By including a correlation matrix of thousands of pariwise tests could inflate false-positive associations without biological meaning and without improving the ecological interpretation of the data.

We therefore choose to maintain an interpretative and hypothesis-generating aspect by only describing the metabolite variation in relation to the broad seasonal environmental data rather than attempting a formal statistical correlation where the current data set is not powered to support.

6 Linking Metabolites to Bioactivity  While both metabolite profiles and bioactivities (e.g., antioxidant capacity, methane inhibition) were measured, the study does not establish a quantitative link between them. The key question—“Which specific metabolite(s) are the main drivers of the observed bioactivities?”—remains unanswered. It is recommended to apply multivariate statistical modeling (e.g., PLS regression) to identify the metabolites that best predict bioactivity.

Response: We thank the reviewer for this comment. However, we would like to clarify that no direct bioactivity measurements were performed in this study. The experimental design focused exclusively on untargeted metabolomic profiling and compositional characterization (dry matter, ash, total polyphenols) of Fucus serratus and F. vesiculosus across a full annual cycle. References to bioactivity or methane inhibition in the text are contextual, highlighting potential applications of the studied metabolites based on existing literature, not results obtained here.

We have added the following statement in the Conclusions: “Although this study did not evaluate bioactivity parameters directly, the observed tem-poral patterns and metabolite identities provide a biochemical basis for future targeted evaluations of antioxidant or other bioactivity potential.”

7 Formatting issues   Journal names in the Reference list should be consistently abbreviated. Latin binomials (including those in the main text) should be italicized. Additionally, some references (e.g., Ref. 38) are missing page numbers or other required details.

Response: Thank you for this observation. We have reviewed the entire reference list and the main manuscript text to ensure consistent use of abbreviated journal names according to MDPI style, and all Latin binomials in the main text and figures and references are now italicized.

Reviewer 2 Report

Comments and Suggestions for Authors

This manuscript presents an interesting study on the Metabolomic insights into the phytochemical profiles and seasonal shifts of two species of Fucus, F. serratus and F. vesiculosus harvested in one site in Denmark. I find the manuscript interesting and, in my opinion, the work deserves to be published in an international journal such as Marine Drugs. I therefore consider that this manuscript could be accepted for publication, after further improvements are made. Here are some ideas for improvement:

The title would benefit from being more specific, as ‘in Nordic waters’ implies multiple sites... yet only one site was sampled in the study

Line 36. Other articles have been studied on these two species along the Atlantic coast.

Not just this article 1.List other articles that have worked on these two species

Line 39. The same applies to applications related to these two species of Fucus... there are loads of articles on this subject!

End of the introduction. Given that the authors then focus on the metabolites produced by the two species of Fucus, it would be interesting to mention in the introduction what is known about the metabolites produced by species of the genus Fucus,

in particular phenolic compounds (PC)... which will be measured later... There is an article by Catarino that presents the PC of Fucales. Catarino et al (2022), which could be added together with a paragraph on what is known in both species.

Lines 78-81. As written, one wonders whether the study covered several sites?

Line 113. Put Spirorbis in italics

Line 126. Materials and methods. Problem with quantifying CP in brown algae. Why not use phloroglucinol? Because gallic acid is absent or is not the dominant CP in brown algae...

Line 134. At the end of the sentence, cite studies to support the idea mentioned here about photoprotection

Line 140. Has this already been demonstrated in Fucus? In other species of Fucales?

Figure 3. Do not use solid lines to connect the dots. What does a biological replica represent? An individual? Or part of an individual? It would be interesting to link the variations observed with environmental parameters. Is there a connection? Fv has seasonal production (peaking in August), while Fs shows high TPC fluctuations during the end spring/summer and lower TPCs the rest of the year. It would be interesting to discuss this. Has this already been demonstrated? Cite articles about that… See Connan et al. 2004 + other studies

Figures 4&5. I am surprised that there is no overlap between Fs and Fv, as they are species of the same genus. They should therefore have some characteristics in common... However, regardless of the sampling month, during the annual monitoring, there is no overlap between months... Is the chemistry of each species therefore different? The authors should discuss this point. In the legend it is noted metabolomic shifts. What do metabolomic shifts represent? ESI+ MS spectrum signals. What data? To be specified in the caption, as the figure should be understandable by reading the caption...

Line 172. Given the results and discussion section, the results must be compared with those in the literature. Have similar or different results been highlighted in the literature?

Figure 6. This figure is very interesting. Please note that some compound names are truncated. What is the order in which the compounds appear? Some are positive, others negative, others are not mentioned, or their names are truncated...

Log2 Abundance could have been put on the same scale to enable comparison of the compounds. It would have been interesting to classify the compounds: carbohydrates, peptides, phlorotannins, etc., or to group them by compound: similar, different between the two species.

Line 194. Line 194. What does this number correspond to? It mentions log2 FC... What does FC stand for? Because on the graphs, it is labelled abundance. Two unidentified compounds are noted. The mass of the compounds has not been obtained??

Line 205. What does antifouling protection have to do with it? Since it wasn't mentioned before, I'm wondering

Line 208. Be careful when writing sulphate/sulfate. Use only one spelling and be consistent. Other places in the article show both spellings

Lines 210–214. Use Catalino et al. (2022), which presents the phlorotannins of Fucales, the order to which Fucus species belong

Lines 223-225. The problem with the dosage, which uses gallic acid instead of phloroglucinol!!! As a result, the phenolic compound content is underestimated!

Lines 254-257. It would be interesting to develop these results further. FS and FV are not positioned in the same way on the rocky foreshore: FS is below FV. Could the sulphate level be a means of adapting to the emersion time? There are articles on this subject in the literature, focusing on sulphated fucans, but could other compounds play this role?

Line 262. One species is seasonal (FV) while the other has more spread-out maximum values (FS) ... it would be interesting to discuss these two species

Figure 7. Not clear. Is this a molecular network? If so, would it be interesting to show it for each species of Fucus?

Figure 8. I do not understand the colour coding of the scale: 1-2 in red ; 0 -1 to -2 in blue. To be explained in the legend

Figure 11. Another way of presenting the results: dendrogram associated with the heatmap. Why not add a dendrogram to the previous clusters?

Line 442 + 599, as well as numerous bibliographical references. Put the genus and species names in italics

Line 443. Only one site was sampled over the course of a year. Why does the title give an overview of Nordic waters?

Author Response

Reviewer 2

Comments and Suggestions for Authors

This manuscript presents an interesting study on the Metabolomic insights into the phytochemical profiles and seasonal shifts of two species of Fucus, F. serratus and F. vesiculosus harvested in one site in Denmark. I find the manuscript interesting and, in my opinion, the work deserves to be published in an international journal such as Marine Drugs. I therefore consider that this manuscript could be accepted for publication, after further improvements are made. Here are some ideas for improvement:

The title would benefit from being more specific, as ‘in Nordic waters’ implies multiple sites... yet only one site was sampled in the study

Response: We appreciate the reviewer’s observation. The title has been modified to more accurately reflect the sampling scope and avoid overgeneralization.

Line 36. Other articles have been studied on these two species along the Atlantic coast.

Not just this article 1.List other articles that have worked on these two species

Response: We have now included additional references covering studies on F. serratus and F. vesiculosus across the Atlantic coast.

Line 39. The same applies to applications related to these two species of Fucus... there are loads of articles on this subject!

Response: We have expanded this section and cited relevant studies illustrating the bioactive and applied potential of Fucus spp. Please see references 5-8 plus the several cited papers from Catarino et al.

End of the introduction. Given that the authors then focus on the metabolites produced by the two species of Fucus, it would be interesting to mention in the introduction what is known about the metabolites produced by species of the genus Fucus, in particular phenolic compounds (PC)... which will be measured later... There is an article by Catarino that presents the PC of Fucales. Catarino et al (2022), which could be added together with a paragraph on what is known in both species.

Response: Agreed. A concise paragraph was added to summarize current knowledge of the metabolite classes and phenolic compounds typical of Fucus spp. The paragraph is marked in red.

Lines 78-81. As written, one wonders whether the study covered several sites?

Response: We have clarified this by changing the title and additionally in materials and methods : “All samples were collected from the same site ensuring environmental consistency across the sampling campaign.”

Line 113. Put Spirorbis in italics

Response: Corrected. All mentions of Spirorbis are now in italics.

Line 126. Materials and methods. Problem with quantifying CP in brown algae. Why not use phloroglucinol? Because gallic acid is absent or is not the dominant CP in brown algae...

Response: We acknowledge the reviewer’s valid point. However, we intentionally used gallic acid equivalents (GAE) to enable cross-comparability with prior seaweed antioxidant studies employing the Folin–Ciocalteu method, while we recognize the method’s limitations in quantifying different types of polyphenols.

Line 134. At the end of the sentence, cite studies to support the idea mentioned here about photoprotection

Response: Thank you for this observation. We have added reference for support.

Line 140. Has this already been demonstrated in Fucus? In other species of Fucales?

Response: This has been previously shown by Heavisides aet al. 2018 in Fucus vesiculosus.

Figure 3. Do not use solid lines to connect the dots. What does a biological replica represent? An individual? Or part of an individual? It would be interesting to link the variations observed with environmental parameters. Is there a connection? Fv has seasonal production (peaking in August), while Fs shows high TPC fluctuations during the end spring/summer and lower TPCs the rest of the year. It would be interesting to discuss this. Has this already been demonstrated? Cite articles about that… See Connan et al. 2004 + other studies

Response: We appreciate the reviewer’s careful comments. We chose to retain connecting lines in Figure 3 because they represent temporal continuity and highlight the monthly progression of mean total phenolic content (TPC) across the year, rather than discrete or unrelated time points. Each data point represents an independent biological replicate (n=5 individual thalli). Regarding the observed trends, the seasonal increase in TPC during late spring–summer and its reduction toward winter has been previously described in Fucus spp. – spiralis, serratus, vesiculosus (Cruces et al., 2012). This pattern could reflect photoadaptive and stress-protective accumulation of phlorotannins in periods of elevated irradiance and temperature as described by Connan et al. 2004. We have added some further explanations in the Results and Discussion to clarify this.

Figures 4&5. I am surprised that there is no overlap between Fs and Fv, as they are species of the same genus. They should therefore have some characteristics in common... However, regardless of the sampling month, during the annual monitoring, there is no overlap between months... Is the chemistry of each species therefore different? The authors should discuss this point. In the legend it is noted metabolomic shifts. What do metabolomic shifts represent? ESI+ MS spectrum signals. What data? To be specified in the caption, as the figure should be understandable by reading the caption...

Response: We thank the reviewer for these observations. Although Fs and Fv belong to the same genus, they occupy distinct ecological niches within the intertidal gradient: Fv typically dominates higher, more emersed zones, while Fs is found in lower, subtidal zones. These differences in light exposure, desiccation stress, and salinity fluctuation strongly influence their metabolic regulation and lead to distinct metabolomic fingerprints. The observed absence in overlap (as the reviewer points out in multivariate space) is yes, a reflection of species-specific metabolomic figerprints. In fact, many core metabolites (e.g., common sugars, amino acids) are shared but contribute less to the variance structure driving separation. We have modified the figure caption to be better understandable. Previous (now removed) word of “metabolic shifts”, was vague and misleading. But when we discuss metabolic shifts we are specifically looking at changes in the intensity of metabolites measured using LC-HRMS.

Line 172. Given the results and discussion section, the results must be compared with those in the literature. Have similar or different results been highlighted in the literature?

Response: A GC-MS study was performed by Rickert et al 2016, focusing on seasonal variations of the surface metabolome of FS and FV. They also have reported large differences between spring/summer and autumn/winter months. We have added to the discussion.

Figure 6. This figure is very interesting. Please note that some compound names are truncated. What is the order in which the compounds appear? Some are positive, others negative, others are not mentioned, or their names are truncated...

Log2 Abundance could have been put on the same scale to enable comparison of the compounds. It would have been interesting to classify the compounds: carbohydrates, peptides, phlorotannins, etc., or to group them by compound: similar, different between the two species.

Response: Thank you for observing these inconsistencies. We have corrected the figure so that the names are not truncated. Because of the large number of compounds and large differences between scales (and abundance) it creates a very illegible figure with very small groups and very difficult to interpret. We have left it as it is. The “pos/neg” is meant to indicate the polarity in which these metabolites were found, all compounds have this information. We have modified the figure legend to be easier to follow. The current order of the compounds are by absolute log₂ fold change (|log₂FC|) between species which makes it a lot easier to see the features that are strongly different from FS to FV irrespective of collection month.

Line 194. Line 194. What does this number correspond to? It mentions log2 FC... What does FC stand for? Because on the graphs, it is labelled abundance. Two unidentified compounds are noted. The mass of the compounds has not been obtained??

Response: We have clarified in the new figure legend description (FC stand for fold change). The two unidentified compounds were removed from the re-made figure, mainly because we want to keep only known and well characterized metabolites. All compounds, including unknowns have mass/charge and retention time information (found in supplementary) however the databases are limited and could only suggest a compound class for those specific two.

Line 205. What does antifouling protection have to do with it? Since it wasn't mentioned before, I'm wondering

Response: We refer to fouling as the presence of Spirorbis which was seen in higher with FV compared to FS.

Line 208. Be careful when writing sulphate/sulfate. Use only one spelling and be consistent. Other places in the article show both spellings

Response: Thank you very much for this. We have made corrections throughout the text to maintain consistency.

Lines 210–214. Use Catalino et al. (2022), which presents the phlorotannins of Fucales, the order to which Fucus species belong

Response: Thank you we have included.

Lines 223-225. The problem with the dosage, which uses gallic acid instead of phloroglucinol!!! As a result, the phenolic compound content is underestimated!

Response: We fully agree that the Folin–Ciocalteu assay using gallic acid does not directly quantify the absolute phlorotannin concentration in brown algae, as phloroglucinol would be a more structurally representative standard. However, the use of gallic acid is a widely established convention that allows cross-comparison among studies using the same colorimetric approach for total phenolic content (TPC), including many on Fucus spp.. The intent of our measurement was to quickly compare relative seasonal and interspecific trends rather than to obtain absolute phlorotannin concentrations for which TPC would also probably not be the best method. We have added a few more lines regarding this in the discussion.

Lines 254-257. It would be interesting to develop these results further. FS and FV are not positioned in the same way on the rocky foreshore: FS is below FV. Could the sulphate level be a means of adapting to the emersion time? There are articles on this subject in the literature, focusing on sulphated fucans, but could other compounds play this role?

Response: We thank the reviewer for this valuable suggestion. Indeed FV, which is more frequently emersed, could experience higher osmotic and desiccation stress, potentially requiring enhanced production of sulfated or osmoregulatory compounds. However, our untargeted LC–HRMS analysis show several sulfate-containing metabolites (e.g., choline sulfate, aromatic acid sulfates) were more abundant in FS, while other specific sulphated compound were higher in FV, therefore difficult to draw precise conclusions.

Line 262. One species is seasonal (FV) while the other has more spread-out maximum values (FS) ... it would be interesting to discuss these two species

Response: We thank the reviewer for the comment. However, it is not fully clear what the reviewer refers to. In our results, both FS and FV show clear seasonal variation in their metabolomic profiles. Only the total phenolic content (TPC) displays a more pronounced summer maximum, but this assay is broad and not specific to individual metabolites. The untargeted metabolomics data show fluctuations throughout the year for both species, so we do not observe that F. vesiculosus is more “seasonal” than F. serratus.

Figure 7. Not clear. Is this a molecular network? If so, would it be interesting to show it for each species of Fucus?

Response: Figure 7 is not a molecular network, but rather a time-series clustering analysis performed using the Mfuzz algorithm. This method groups metabolites with similar temporal abundance profiles (based on standardized z-scores) across the 12 sampling months. Each panel (clusters 1–6) represents a distinct temporal pattern of metabolite abundance shared by F. serratus (FS) and F. vesiculosus (FV). The analysis highlights how groups of metabolites co-vary through the year and differ between species in their seasonal regulation. We have revised the figure caption to better clarify this.

Figure 8. I do not understand the colour coding of the scale: 1-2 in red ; 0 -1 to -2 in blue. To be explained in the legend

Response: The colour scale represents the standardized abundance (z-score) of each metabolite, where red indicates higher relative abundance and blue indicates lower relative abundance within each species over time. The range (–2 to +2) reflects the standardized deviation from the mean abundance after z-score normalization. The figure legend has been revised to explicitly explain this coding.

Figure 11. Another way of presenting the results: dendrogram associated with the heatmap. Why not add a dendrogram to the previous clusters?

Response: We thank the reviewer for the suggestion. However, the clustering shown in this figure was performed using the Mfuzz algorithm, which applies a soft (fuzzy) time-series clustering approach rather than hierarchical clustering. As such, the output does not generate a dendrogram structure — each metabolite can belong to more than one temporal cluster with varying membership scores. Adding a dendrogram would therefore not accurately represent the underlying clustering method or the temporal dynamics of the data.

Line 442 + 599, as well as numerous bibliographical references. Put the genus and species names in italics

Response: Thank you very much! We have corrected all references.

Line 443. Only one site was sampled over the course of a year. Why does the title give an overview of Nordic waters?

Response: Yes, as mentioned in the beginning and introduction we have corrected and modified title and materials and methods.

Reviewer 3 Report

Comments and Suggestions for Authors

Dear authors, this paper apporach a very interesting topic, i have few suggestion:

Introduction: Please contextualize the work by reviewing prior metabolomics studies on other brown algae; if none exist, highlight the pioneering nature of this study.

Materials and Methods: Provide full details of all instrumentation used. Include more details of chromatographic parameters 

Author Response

Reviewer 3:

Dear authors, this paper apporach a very interesting topic, i have few suggestion:

Introduction: Please contextualize the work by reviewing prior metabolomics studies on other brown algae; if none exist, highlight the pioneering nature of this study.

Response: Thank you very much for your comments and suggestions. We have not found a lot of studies doing untargeted metabolomics on Fucus, the ones that are available we have included in the results/discussion and where needed to be pointed out.

Materials and Methods: Provide full details of all instrumentation used. Include more details of chromatographic parameters.

Response: the materials and methods section contain all the information necessary for accurate sample preparations and replication of the instrumental analysis and data analysis.All revisions points mentioned in the pdf document have been corrected or improved based on the comments.

Reviewer 4 Report

Comments and Suggestions for Authors

Dear authors

Excellent work done by you. Some sections need to be improved, especially the references. In that section you will see that what needs to be corrected is highlighted in orange, whether it means abbreviating the name of the journal, writing the scientific names in italics, or standardizing the citation method.
Please improve the figures and address the comments I wrote in the PDF. It's important to highlight the scope of your research.
Best regards

Author Response

Reviewer 4:

Excellent work done by you. Some sections need to be improved, especially the references. In that section you will see that what needs to be corrected is highlighted in orange, whether it means abbreviating the name of the journal, writing the scientific names in italics, or standardizing the citation method.
Please improve the figures and address the comments I wrote in the PDF. It's important to highlight the scope of your research.
Best regards

Response: Thank you very much for your comments and suggestions. The reference style is MDPI specific, so we apologize for any issues. But we have hopefully corrected all issues related to citations.
